# Antimicrobial resistance and virulence analysis of *Escherichia coli* carried by black-faced spoonbill (*Platalea minor*) in Liaoning, China

Bing Liang[1,2], Bo-wen Jiang[1,2], Yue Yuan[3], Tie-cheng Wang[1,2], Yuan-guo Li[1,2], Ling-wei Zhu[1,2], Jun Liu[1,2], Xue-jun Guo[1,2], Xue Ji[1,2]*, Yang Sun[1,2]*

**1** Changchun Veterinary Research Institute, Chinese Academy of Agricultural Sciences, Changchun, China, **2** Key Laboratory of Jilin Province for Zoonosis Prevention and Control, Changchun, China, **3** Shenyang 35th Middle School, Shenyang, China

* sunyang10@hotmail.com (YS); ji_xuecn@aliyun.com (XJ)

## Abstract

To better protect the black-faced spoonbill and explore the transmission and public health risks of drug-resistant pathogenic bacteria carried by this rare migratory bird, 249 fecal samples were collected from the black-faced spoonbills on Shicheng Island, Zhuanghe City, Liaoning Province, China in 2017 and *Escherichia coli* (*E. coli*) were isolated and identified. The BD Phoenix™-100 automated microbial identification system and E-test strips were used to determine the resistance phenotype and minimum inhibitory concentration (MIC) value of the isolates. Whole genome sequencing and analysis were conducted on multidrug resistant (MDR) isolates and an evolutionary tree was constructed. Seventy-four samples had *E. coli* isolated (29.7%), 43.2% of isolates carried an antimicrobial resistant phenotype and 21.6% isolates were considered to be MDR strains. The tetracycline resistance rate was highest (41.9%), followed by ampicillin (20.3%), chloramphenicol (18.9%) and trimethoprim/sulfamethoxazole (17.6%). The highest MIC values of ampicillin, cefazolin, cefotaxime, ciprofloxacin, levofloxacin, and chloramphenicol were eight to 32 times the cutoff point for antimicrobial resistance. Twenty-one types of resistance genes and 35 virulence-associated genes were detected in the MDR isolates and the main type of plasmid was IncFIB. The *mcr-1* gene was detected for the first time in MDR *E. coli* isolated from the black-faced spoonbill. The *astA* gene was only detected in nine isolates with undetected antimicrobial resistant phenotype. Genetic diversity analysis showed that nine *E. coli* isolates were mainly concentrated and clustered in independent evolutionary branches in the phylogenetic tree, with serotypes of O-:H8 and the multilocus sequence typing was ST2077, which were in the same evolutionary branch as habitat domesticated animals and environmental isolates. In summary, *E. coli* may serve as a reservoir of resistance and virulence genes in migratory birds and may be transmitted to other species during migration, with virulent or multidrug resistant *E. coli* a potential threat to the rare black-faced spoonbill and to human public health.

**Data availability statement:** All relevant data are within the paper and its Supporting information files.

**Funding:** The authors work at the Changchun Veterinary Research Institute, Chinese Academy of Agricultural Sciences, Key Laboratory of Jilin Province for Zoonosis Prevention and Control. Our institution's funding is provided by the National Key Research and Development Program of China (Grant No. 2023YFF1305401) and the Wildlife Epidemic Monitoring Project of the State Forestry and Grassland Administration (Grant No. [2023] 58). The funders had no role in study design, data collection and analysis, decision to publish, or preparation of the manuscript.

**Competing interests:** No conflict of interest exits in the submission of this manuscript, and all the authors listed have reviewed the final version of the manuscript and approve it for publication.

## Introduction

The black-faced spoonbill (*Platalea minor*) is a medium-sized wading bird belonging to the Threskiornithidae family and the Platalea genus, which commonly inhabits inland lakes, estuaries, reed swamps and other areas. It feeds on fish, shrimp, and crabs, as well as mollusks and crustaceans and is presently mainly found in nine countries, including South Korea, North Korea, Russia, Cambodia, and China. In China, most black-faced spoonbills live and breed on the bird island of Shicheng, Zhuanghe City and Liaoning Province in the summer and migrate to southern China in the winter. Every November, they migrate along the eastern coast of China, passing through Sushan Island in Shandong, Yancheng in Jiangsu, Chongming Island in Shanghai and Zhoushan Islands in Zhejiang, to pass the winter in Fujian and Taiwan. In February of the following year they migrate to Liaoning and arrive in Shicheng Township between March and April to start nesting and breeding [1]. In the 1830s, with increasing water pollution and habitat destruction, the population decreased sharply and by the 1880s, the population had decreased to fewer than 300 birds [2]. In 1989, the black-faced spoonbill became a globally endangered species and was listed as a second level protected wild animal in China's National Key Protected Wildlife List. In 2021, it was elevated to a first level protected wild animal. Currently, there are only a thousand black-faced spoonbills in the world [2,3].

Members of the class Aves occupy a wide range of niches throughout the world and trillions of microbes live in the intestinal tracts of birds. Through high-throughput sequencing technology, Cao Jian found that the gut bacterial communities in migratory bird populations were distinct from each other, with *Pseudomonas*, *Peptostreptococcaceae nomame* and *Escherichia* shared by all species [4].

*Escherichia coli* (*E. coli*) is one of the most common and important pathogenic bacteria in birds [5] and mainly causes diarrhea in animals including humans. According to the differences in priority colonization sites, the virulence factors and corresponding clinical signs of diarrheal *E. coli* in the host, it can be divided into enteropathogenic *E. coli* (EPEC); enterohemorrhagic *E. coli* (EHEC/STEC); enteroaggregative *E. coli* (EAEC); enterotoxic *E. coli* (ETEC); enteroinvasive *E. coli* (EIEC); diffusely adhesive *E. coli* (DAEC) and a newly identified type of adhesive invasive *E. coli* (AIEC) [6]. The type of *E. coli* that causes disease in birds is termed avian pathogenic *E. coli* (APEC), which is a subgroup of extraintestinal pathogenic *E. coli* (ExPEC) and can be associated with high mortality [7].

The long-term use and misuse of antibiotics has increased the resistance of pathogenic bacteria to antibiotics and the emergence of drug-resistant bacteria has caused serious agricultural, animal and human health problems worldwide [8]. Horizontal transmission of antimicrobial resistance genes (ARGs) is one of the main ways by which bacteria obtain drug resistance and plays an important role in the development of bacterial multidrug resistance (MDR) [4]. Given that *E. coli* is considered to be one of the most important bacteria responsible for horizontal transfer of ARGs, drug-resistant *E. coli* is of public health concern [9].

Migratory birds are a natural reservoir of drug-resistant bacteria as they are globally abundant in number and have a wide range of activity. They can carry foreign

ARGs during migration, which helps to produce MDR bacteria and spread ARGs in the environment [6] and through monitoring, the cross transmission of pathogenic bacteria among wild animals, humans, and livestock has been well characterized [6,10,11].

Research is limited on *E. coli* carried by black-faced spoonbills worldwide. In this study, a combination of polymerase chain reaction (PCR) and second generation sequencing (SGS) technology was used to verify the results of the minimum inhibitory concentration (MIC) test, with the aim of investigating the antimicrobial resistance spectrum of *E. coli* isolates from Liaoning Black-faced spoonbills, the drug resistance genes involved, the correlation and distribution of integrase and drug resistance genes and the genetic relationship of *E. coli* isolates, using SGS technology.

## Materials and methods

### Sample collection

A total of 249 fecal samples from black-faced spoonbills were randomly collected from Shicheng Township, Zhuanghe City, Liaoning Province from May to June 2017. All samples were collected under the supervision of the Wild Animal Sources and Diseases Inspection Station, National Forestry and Grassland Bureau of China and did not cause any harm to the animals. The swabs were placed in 1 mL physiological saline at 4°C for a short time and transported to the laboratory on ice for further processing.

### Isolation and identification of *E. coli*

Each sample in physiological saline was vortexed and the mixture was left to stand for 5 min at room temperature. The samples were cultured on MacConkey agar (QingDao Hope Bio), with large, flat, red single colonies with dark red centers two to three mm in diameter assumed to be *E. coli* and subjected to secondary purification on MacConkey agar. The strains were subcultured on brain heart infusion (BHI) agar plates, incubated for 16–18 h at 36 ± 1°C and stored at 80°C in BHI broth with 20% glycerol.

### Antimicrobial susceptibility testing

The identification and antimicrobial susceptibility of isolated strains were determined using the NMIC/ID 4 panel of the BD Phoenix™-100 automated microbial identification system (Becton Dickinson, USA). The antibiotics studied included amikacin, gentamicin, imipenem, meropenem, cefazolin, ceftazidime, cefotaxime, cefepime, aztreonam, ampicillin, piperacillin, amoxicillin–clavulanate, ampicillin–sulbactam, piperacillin–tazobactam, colistin, trimethoprim–sulfamethoxazole, chloramphenicol, ciprofloxacin, levofloxacin, moxifloxacin and tetracycline. The determination of the drug sensitivity results was based on the standards of the American Association for the Clinical and Laboratory Standards Institute (CLSI) [12].

### Determination of the minimum inhibitory concentration (MIC)

With reference to the CLSI (2017) enterobacteriaceae susceptibility test technology, the standard E-test strip (Liofilchem, Italy) was used for antimicrobial susceptibility testing on *E. coli* with a drug resistance phenotype. The zone of inhibition was examined to determine the MIC range according to the CLSI guidelines. The MIC breakpoints of each antibiotic for *E. coli* and the range of concentration detected by E-test are shown in Table S1 in S1 File and *E. coli* ATCC25922 was used as the quality control strain.

### DNA extraction

The bacterial cells were recovered from 1 mL pure culture of *E. coli* isolates grown at 37°C for 18 hours, resuspended with an equal volume of ddH$_2$O, boiled for 7 min and centrifuged at 12,000 rpm for 1 min. The recovered supernatant was used for detection of virulence and drug resistance genes.

## Detection of virulence genes by PCR assay

Previously reported primers were used for the detection of the virulence genes of four pathogenic *E. coli* strains, listed in Table S2 in S1 File and the STEC, EPEC, and EAEC were identified by detecting the virulence genes *eae*, *Stx1*, *Stx2*, *LT*, *STa*, *STb*, *ST*, and *astA*. The reaction mixtures were dispensed into 96-well PCR tubes (Beijing Dingguo Changsheng Biotechnology Co. Ltd., China) with 25 μL in each well, containing 1 μL DNA template, 12.5 μL 2×Taq DNA Master Mix (CWBio, Beijing, China), 0.5 μL of each 10 μM primer and 10.5 μL ddH$_2$O. The *E. coli* isolates used as positive controls were CVCC216 as a positive control for *eae*, *ST*, *Sta*, and *Stb*; CVCC216 as a positive control for *LT* and *EAST-1* and EDL933 as a positive control for *Stx1* and *Stx2*.

## Detection of ARGs and integrons

The PCR reactions were used to detect drug resistance genes of *E. coli* isolates, including ESBL genes bla$_{CTX-M}$, bla$_{CTX-M}$ genotype group 1, 2, 9 and bla$_{TEM}$, tetracycline resistance genes *tet (A)*, *tet (B)*, *tet (C)*, *tet (D)*, *tet (M)* and *tet (W)*, sulfonamide resistance genes *sul1*, *sul2*, *sul3*, and *sulA*, chloramphenicol resistance genes *cat1*, *cmlA*, and *floR*, the colistin resistance gene *mcr-1*, and integrase genes *intI1*, *intI2*, *and intI3*. The PCR system and conditions were as described by Yue Yuan et al. [6].

## Whole genome sequencing and bioinformatics analyses

The DNA was extracted from MDR *E. coli* isolates using a bacterial DNA extraction kit (Omega Bio Tek, USA) and high-throughput sequencing of DNA was performed on the Illumina (USA) platform using NovaSeq PE150. For whole-genome sequence analysis, the phylogenetic tree based on single nucleotide polymorphisms (SNPs) was constructed using RealPhy (CH) [13] and observed using the Interactive Tree Of Life (iTOL v6, https://itol.embl.de/) [14]. Antimicrobial resistance genes, virulence genes, plasmids, sequence type (ST) and serotypes were identified using ResFinder 4.1, VirulenceFinder 2.0, PlasmidFinder 2.1, and Serotype Finder 2.0 on the Computable General Equilibrium (CGE) website [14,15]. Isolates of *Escherichia* were assigned to phylogroups using ClermonTyping v1.4.1 (USA) [16].

## Data availability

The *E. coli* genomes sequenced in this study have been uploaded to the National Center for Biotechnology Information (NCBI) Sequence Read Archive (SRA) database associated with BioProject under the accession numbers given in Table S3 in S1 File. The draft genomes of 40 *E. coli* strains distributed in poultry, cows, mink, pigs, migratory birds, cormorants and environmental sources from Japan and Guangzhou, Zhaoqing, Futian, Zhejiang, and Liaoning provinces in China were also downloaded from NCBI and included in the phylogenetic analysis, in Fig 3.

## Results

### Isolation and resistance phenotypes of *E. coli*

Seventy-four (29.72%) *E. coli* strains were isolated with the MDR bacteria resistant to three or more types of antibiotic. Among the isolates, 32 (43.24%) strains were drug-resistant, 16 (21.62%) strains showed multiple drug resistance and only one strain 826red was of the ESBL phenotype. Among the MDR strains, there were six patterns of multiple drug resistance, with TET-AMP-SXT-CHL-GEN accounting for 56.25%, as shown in Table 1. With regard to the antimicrobial resistance rate, tetracycline showed the highest resistance rate (96.88%), followed by AMP, CHL, SXT, GEN and PIP, with rapidly decreasing resistance rates between 20.27% and 13.51%. The resistance rates of the remaining seven antibiotics were below 7%, as shown in Fig 1.

**Table 1. Antimicrobial resistance spectra of strains.**

| strains | Resistance phenotypes | MIC (µg/mL) | Resistance genes | integrons |
|---|---|---|---|---|
| 831 | TET | 32 | *tetA* | – |
| 849 | TET | 64 | *tetA,cmlA* | – |
| 860 | TET | 64 | *tetA* | – |
| 989 | TET | 96 | *tetA* | – |
| 996 | TET | 64 | *tetA* | – |
| 1002 | TET | 64 | *tetA* | – |
| 1006 | TET | 48 | *tetA,sul1* | *intl1* |
| 1012 | TET | 2 | – | – |
| 1030 | TET | 48 | *tetA,floR* | – |
| 1031 | TET | 96 | *tetA,cmlA* | – |
| 1040 | TET | 48 | *tetA,sul2,floR* | – |
| 1047 | TET | 96 | *tetA,cmlA* | – |
| 1053 | TET | 64 | *tetA,cmlA* | – |
| 1063 | TET | 48 | *tetA,sul2,floR* | – |
| 948 | AMP PIP | >256−16 | *bla*$_{TEM-1}$,*tetA,sul2,floR* | – |
| 1024 | TET AMP | 24−12 | *bla*$_{TEM-1}$,*tetA,cmlA* | *intl1* |
| 970 | TET SXT CST | 2-0.094-1 | *tetA,cmlA* | – |
| 937 | TET SXT CST | 48- > 32- > 8 | *tetA,sul2,cmlA, mcr-1* | *intl1* |
| 958 | TET CHL CST | 1.5-16-2 | – | – |
| 830red | TET AMP PIP CHL | 64- > 256−64- > 256 | *bla*$_{TEM-1}$,*tetA,sul2,floR* | – |
| 1018 | TET AMP PIP CHL | 64- > 256−64- > 256 | *bla*$_{TEM-1}$,*tetA,sul2,floR* | – |
| 822red | TET AMP SXT CHL GEN | 48−32- > 32- > 256−48 | *tetA,sul2,floR* | *intl1* |
| 837red | TET AMP SXT CHL GEN | 32−48- > 32- > 256−32 | *tetA,sul2,floR* | *intl1* |
| 848red | TET AMP SXT CHL GEN | 48−48- > 32- > 256−48 | *tetA,sul2,floR* | *intl1* |
| 883 | TET AMP SXT CHL GEN | 48−64- > 32- > 256−48 | *tetA,sul2,floR* | *intl1* |
| 887 | TET AMP SXT CHL GEN | 48−48- > 32- > 256−64 | *tetA,sul2,floR* | *intl1* |
| 890 | TET AMP SXT CHL GEN | 64−48- > 32- > 256−48 | *tetA,sul2,floR* | *intl1* |
| 892 | TET AMP SXT CHL GEN | 32−32- > 32- > 256−48 | *tetA,sul2,floR* | *intl1* |
| 926 | TET AMP SXT CHL GEN | 48−64- > 32- > 256−48 | *tetA,sul2,floR* | *intl1* |
| 984 | TET AMP SXT CHL GEN | 64−32- > 32- > 256−48 | *tetA,sul2,floR* | *intl1* |
| 994 | TET AMP PIP SXT CHL CIP LVX GEN | 48- > 256−24- > 32- > 256- > 32- > 32−32 | *tetA,sul1,sul2,floR,cmlA* | – |
| 826red | TET AMP PIP SAM SXT CHL CIP LVX CFZ CTX | 64- > 256- > 256−64- > 32- > 256- > 32−24- > 256- > 32 | *CTX-M9 group, bla*$_{TEM-1}$,*tetA,sul2,floR* | *intl1* |

Abbreviations: AMP, ampicillin; AZT, aztreonam; CHL, chloramphenicol; CIP, ciprofloxacin; CFZ, cefazolin; CPM, cefepime; CTX, cefotaxime; GEN, gentamicin; LVX, levofloxacin; PIP, piperacillin; SAM, ampicillin–sulbactam; SXT, trimethoprim–sulfamethoxazole; TET, tetracycline.

## MIC of MDR *E. coli*

The maximum MIC values of different antibiotics for 16 *E. coli* isolates were as follows: the MICs of AMP, PIP, CHL, SAM, GEN, CTX and CFZ exceeded 256 µg/mL, the MICs of SXT and CIP exceeded 32 µg/mL and the MIC of tetracycline was 96 µg/mL, as seen in Table 2.

## Assay of ARGs, virulence genes and integrons

The PCR results of ARGs in drug-resistant *E. coli* showed that the main drug-resistant genes were *tet (A)*, *sul2*, *floR*, and *cmlA*, with positive rates of 93.75% (30/32), 53.15% (17/32), 53.15% (17/32), and 25% (8/32), respectively. Other

**Fig 1. Antimicrobial resistance rates of *E. coli*.**

**Table 2. Minimum inhibitory concentration test results.**

| Antibiotic | S | I | R | MIC range (μg/mL) |
|---|---|---|---|---|
| Tetracycline | 9.68%(3/31) | 0 | 90.32%(28/31) | 96−64 |
| Piperacillin | 20.00%(1/5) | 40.00%(2/5) | 20.00%(1/5) | >256--128 |
| Ampicillin | 0 | 0 | 93.33%(14/15) | >256−32 |
| Ampicillin- Sulbactam | 0 | 0 | 100%(1/1) | >256−32 |
| Trimethoprim-Sulfamethoxazole | 7.70%(1/13) | 0 | 92.31%(12/13) | >32−4 |
| Cefazolin | 0 | 0 | 100%(1/1) | >256 |
| Cefotaxime | 0 | 0 | 100%(1/1) | >256 |
| Gentamicin | 0 | 0 | 100%(10/10) | >256--16 |
| Ciprofloxacin | 0 | 0 | 100%(2/2) | >32−6 |
| Levofloxacin | 0 | 0 | 100%(2/2) | >32−8 |
| Chloramphenicol | 0 | 7.14%(1/14) | 92.86%(13/14) | >256--48 |

drug-resistant genes were $bla_{TEM-1}$, *sul1*, and *mcr-1*, with positive rates of 15.63% (5/32), 6.25% (2/32), and 3.13% (1/32), respectively. Only strain 826 belonging to the CTX-M-9 subgroup contained $bla_{CTX-M}$. Only integron gene *intl1* was detected and the positive rate was 40.63% (13/32) and the positive rate of MDR strains was 68.75% (11/16), as shown in Table 3. The *astA* gene was only detected in non-resistant *E. coli*, with a positive rate of 12.2% (9/74).

## Phylogenetic analysis

With reference to the migration route of the black-faced spoonbill and possible hosts associated with migratory birds, the genome sequences of 40 *E. coli* strains were downloaded from NCBI and were combined with 16 multidrug-resistant

**Table 3. Statistical results of the phylogenetic tree, MLST, plasmid type, and serotype for16 *E. coli* strains.**

| Strain | Plasmid pattern | ST typing | ABD typing | H antigen | O antigen |
|---|---|---|---|---|---|
| 822red | IncFIB; | ST2077 | B1 | H8 | – |
| 848red | IncFIB; | ST2077 | B1 | H8 | – |
| 887 | IncFIB; | ST2077 | B1 | H8 | – |
| 926 | IncFIB; | ST2077 | B1 | H8 | – |
| 892 | IncFIB; | ST2077 | B1 | H8 | – |
| 890 | IncFIB; | ST2077 | B1 | H8 | – |
| 984 | IncFIB; | ST2077 | B1 | H8 | – |
| 837red | IncFIB; | ST2077 | B1 | H8 | – |
| 883 | IncFIB; | ST2077 | B1 | H8 | – |
| 1018 | IncFIB; | ST441 | B1 | H10 | – |
| 830red | IncFIB; | ST441 | B1 | H10 | – |
| 826red | IncFIB; IncX4 | ST2179 | B1 | H9 | O9; O9a |

*E. coli* strains to construct an SNP Array evolutionary tree, as shown in Fig 2. Sixteen *E. coli* were derived from migratory birds along the migration route and 24 *E. coli* from local livestock, poultry and the environment in Liaoning. Thirteen isolates were relatively concentrated and clustered in four independent evolutionary branches including nine strains, two strains, one strain, and one strain, respectively. The remaining three isolates were scattered in different evolutionary branches. Some of the strains from black-faced spoonbill migration routes were close to *E. coli* strains in Liaoning Province. In terms of host correlation, the isolates 826red, 937, and 930 were closely related to the *E. coli* strains isolated from chickens, cow manure and mink, respectively. As shown in Table 3, nine ST types of the isolates were ST2077, which might have been isolated from the same family according to the sampling order, while the B1 group (81.25%) was the main phylogenetic group.

## Antimicrobial resistance analysis

Sixteen MDR *E. coli* isolates were sequenced for whole genome analysis. The antimicrobial resistance genes and types of plasmids were assessed with ResFinder (https://cge.cbs.dtu.dk/services/ResFinder/) using default settings [17] on the CGE website. Twenty-one antimicrobial resistance genes were detected, including four β-lactam resistance genes, five aminoglycoside resistance genes, three quinolone resistance genes, two macrolide resistance genes, two sulfonamide resistance genes, one chloramphenicol resistance gene, one rifampicin resistance gene, one tetracycline resistance gene and two trimethoprim resistance genes, as shown in Fig 3. As shown in Table 3, the main plasmid type was IncFIB (87.5%), which mostly presented in avian pathogens rather than in human and avian symbiotic bacteria.

## Virulence analysis

A total of 35 virulence-associated genes were identified using VirulenceFinder on the CGE website, shown in Fig 4. The *csgA*, *fimH*, *gad*, *hlyE*, *nlpI*, *yehA* and *terC* genes had the highest positive rate (100%). The strain 970 contained up to 30 virulence-associated genes and the minimum number of virulence-associated genes was 10, carried by strain 937. As shown in Table 3, five kinds of H antigen were detected, with H8 the most frequent (56.25%). The O-antigen serotypes were detected in only four strains, including O9 (1/4), O8 (2/4) and O11 (1/4).

## Discussion

Antibiotics are often used to prevent and treat bacterial infection or as growth promoters in livestock and poultry, which may be the main reasons for bacterial resistance to antibiotics [18,19], and the transmission of resistance genes directly

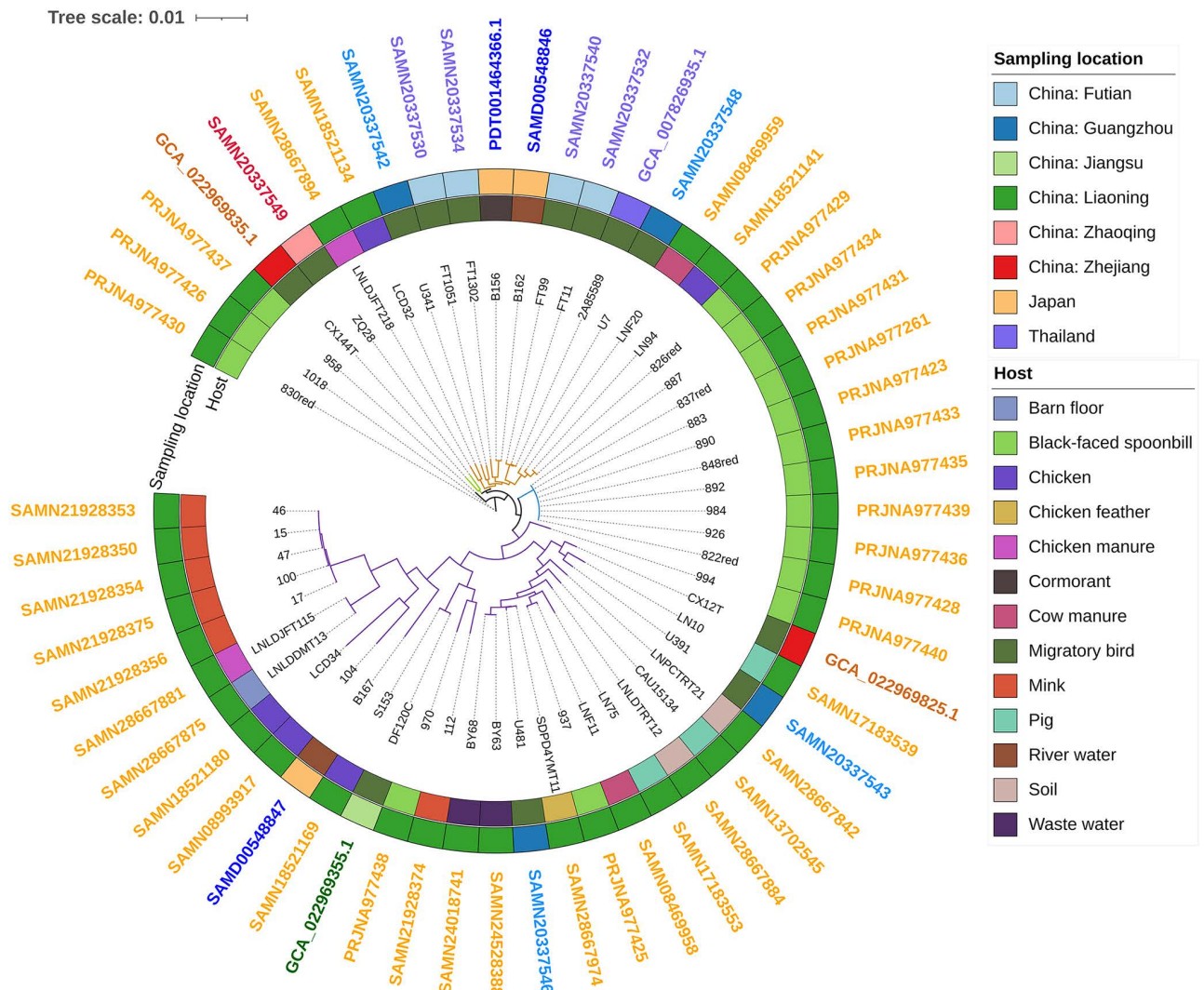

**Fig 2. Single nucleotide polymorphism (SNP) evolutionary analysis of *E. coli* strains.**

threatens human health, wildlife and the environment. In 2019, the Chinese Ministry of Agriculture and Rural Affairs issued Document No. 194, which required feed production enterprises to stop producing commercial feed containing growth promoting drugs and feed additives from July 1, 2020, marking the beginning of the era of non-antimicrobial production in the Chinese feed industry.

In this study, the resistance rate of TET was highest, followed by that of AMP. This resistance pattern is consistent with that of *E. coli* in most clinical samples, or those from livestock and poultry farms [20,21]. This finding reflects the importance of One Health, through which bacteria isolated from wild animals can be monitored indirectly for antimicrobial resistance [22,23].

A systematic study was conducted on the epidemic and molecular characteristics of *E. coli* carried by black-faced spoonbill on Shicheng Island, Liaoning Province. Robé's research shows that *E. coli* can colonize the intestinal tract of birds within 24 hours [24]. The low isolation rate of *E. coli* indicated that they do not exist as symbiotic bacteria in the intestine of black-faced spoonbills as they do in humans and were probably obtained from the environment. Most of the

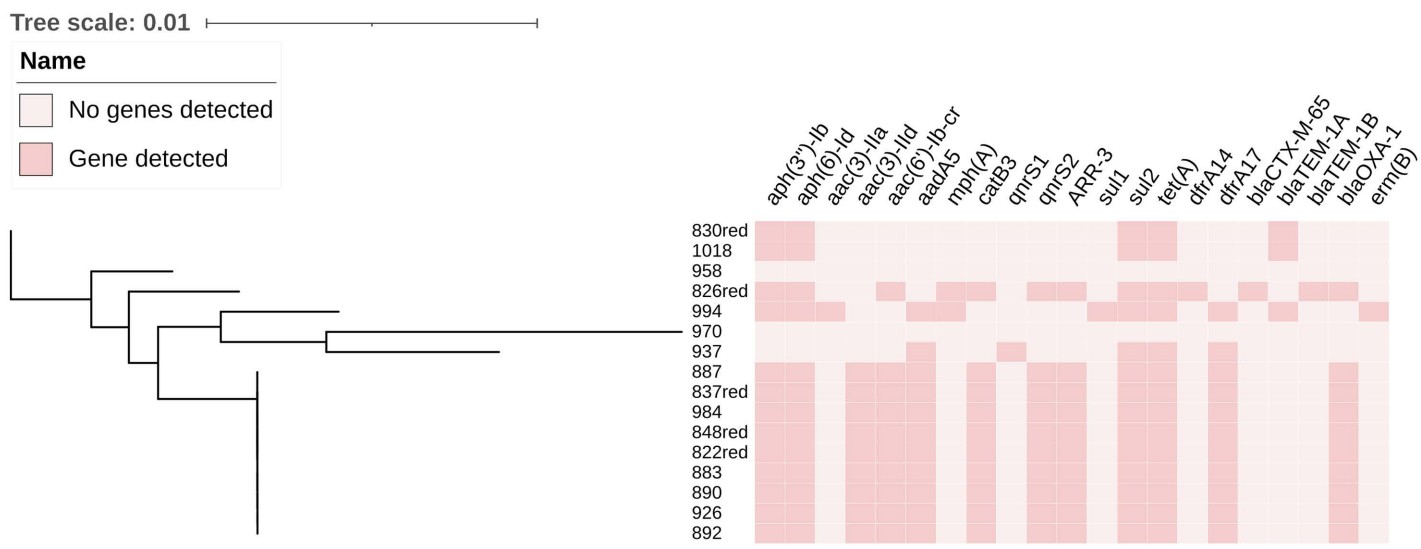

**Fig 3. Analysis of antimicrobial resistance genes in 16 MDR *E. coli* isolates.**

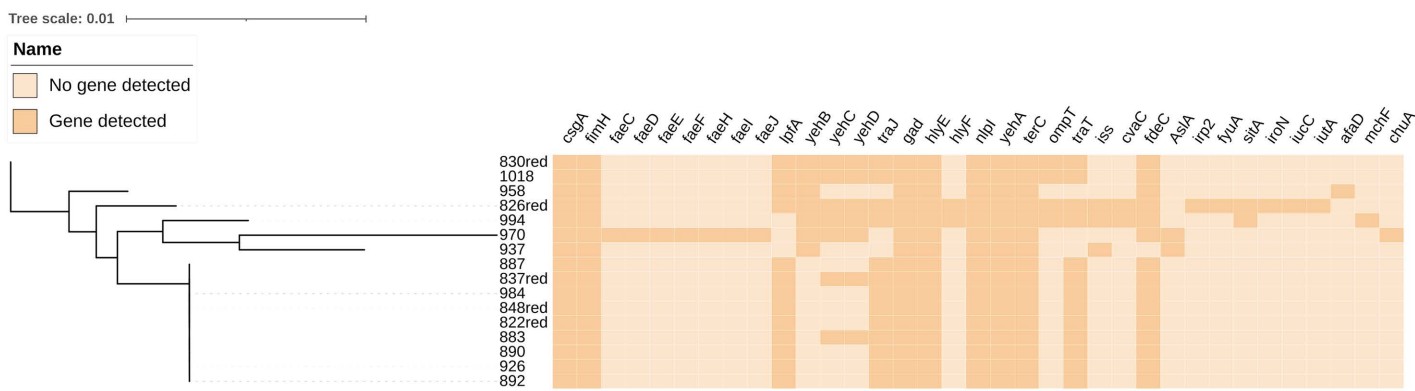

**Fig 4. Analysis of virulence-associated genes.**

black-faced spoonbill's foraging sites are concentrated in fisheries, on mudflats and on the coasts of wintering sites where aquaculture is often located. Given the abundance of offshore aquaculture and the discharge of sewage and wastewater, there is likely to be a certain degree of pollution with antimicrobial residues in the offshore area, which may be a reason for the emergence of antimicrobial resistant bacteria in wild birds. In this study, the percentage of antimicrobial resistant *E. coli* was relatively high, at about 43%. Compared with migrant birds in Inner Mongolia, Ningxia, Jiangxi, and Guangzhou, the antimicrobial resistance of *E. coli* from black-faced spoonbill in Shicheng Island ranked first in five regions [25]. The TET-AMP-SXT-CHL-GEN was the main form of multidrug resistance, accounting for 56.25% and was related to several long-acting antibiotics including TET, AMP and SXT used by the Chinese aquaculture industry in clinical treatment and disease prevention [26,27]. The resistance rate of *E. coli* to AMP, SXT, CHL and GEN ranged from 30 to 50% and that of TET was above 96%. This is not surprising, because TET resistance is the most common type of resistance observed in wild small mammal isolates to date and in farming and aquaculture TET is often used as a first-line antibiotic for food animal disease prevention and growth promotion, so its widespread application predisposes to high resistance rates [28].

The highest MIC values of AMP, CFZ, CTX, CIP, LVX and CHL were eight to 32 times the cutoff point for antimicrobial resistance, showing high levels of antimicrobial resistance in the vertical direction and diverse resistance patterns in the transverse direction. The only ESBL *E. coli* isolate 826red also reflected these characteristics.

Antimicrobial susceptibility testing (AST) is the cornerstone of monitoring the occurrence of antimicrobial resistance (AMR). At present, broth microdilution (BMD) and paper diffusion are considered the gold standards for AST in vitro. In this study, the BD Phoenix™-100 automated microbial identification system and the E-test were used for antimicrobial susceptibility testing, but the results for some isolates were inconsistent between the two methods. Bortolaia showed that, even when using the same method, inconsistent results may be due to bacterial growth, reagent stability, and environmental conditions [29], so more than two methods should be used to improve the accuracy of the results. From a clinical perspective, the detection of AMR genes may be more correlated with drug resistance than the phenotypes observed in vitro and they can be used to avoid initiating inappropriate treatment [29]. The association between the genotype and phenotype of antimicrobial resistance was complicated in this study. Three isolates 937, 970, and 958 exhibited colistin resistant phenotypes, but only strain 937 showed an MIC value exceeding 8 µg/mL and carried the *mcr-1* gene and it also carried three plasmids, IncFIB, IncI1-I, and p0111. Many studies have shown that AMR genes can be transmitted between different bacterial communities through horizontal gene transfer, leading to the high prevalence of antimicrobial resistance genes and the emergence of multidrug resistance. Integrators especially class 1 integrons play an important role in the horizontal transfer of antimicrobial resistance in environmental bacteria [30] and a high frequency of the *intl1* gene was found in resistant isolates [30], so detecting the integrons in *E. coli* isolates is crucial for evaluating the potential of the bacteria to become multidrug-resistant and participate in the horizontal transfer of antimicrobial resistance. In this study, only *intl1* was detected, and the carrier rate in multidrug-resistant bacteria was significantly higher than that in non-drug-resistant bacteria, which was consistent with Yuan Yue's conclusion [25].

In addition to integrators, plasmids are more common mobile elements. The IncFIB plasmid plays a potential role in spreading antimicrobial resistance and virulence genes among various bacteria. In this study, 87.5% (14/16) of MDR *E. coli* isolates carried IncFIB plasmids, indicating that IncFIB plasmids were important to the presence of antimicrobial resistance genes and their spread among different species.

In this study, the isolates mainly carried *tet (A)*, *sul2*, and *floR* resistance genes, consistent with the resistance genotypes found on chicken and pig farms as described by Xue Yang et al. [31]. The PCR results for detection of resistance genes were completely consistent with those of SGS for *sul1*, *sul2*, *tetA*, and bla_{TEM-1}, but Fig 4 showed that SGS was the best technique for detection of drug resistance genes as the details of the resistance genes, such as subtype and location, were more accurate.

Among the MDR isolates, chloramphenicol resistance genes *floR* and *cmlA* were detected in some strains by PCR and the MIC values exceeded 256 µg/mL, but no CHL resistance genes were found by SGS. This may be explained by the high custom similarity of *floR* and *cmlA* during the comparison of SGS results, which is related to the depth of whole genome sequencing (WGS) data reading. There is no clear threshold for the acceptable reading depth of AMR genes and mutations detected in WGS data. While dealing with low reading depth, we defined most of the parameters ourselves, aware that visualizing the reading depth parameter was extremely important when analyzing sequences [29].

The preliminary detection of antimicrobial resistance genes through PCR has certain drawbacks, such as gene mutations and partial sequence loss in the target gene sequence in the natural environment. In addition, poor primer specificity, issues with the PCR reaction system, inaccurate DNA polymerase, and Mg$^{2+}$ concentration can all affect the PCR detection results, so SGS technology can give a better and more comprehensive understanding of bacteria carrying resistance and virulence genes.

In this study, the pathogenic types of 74 *E. coli* strains were determined based on the virulence factors carried by the pathogenic *E. coli*. The EAST1 virulence gene was detected in only nine strains, which belonged to the enteroaggregative *E. coli* (EAEC). These strains did not have antimicrobial resistance, further confirming the negative correlation between

some virulence genes and drug resistance [14]. A total of 35 virulence factors (Fig 4) were identified in 16 MDR *E. coli* strains. The *csgA-fimH-tpfA-traJ-gad-hlyE-nlpI-yehA-terC-traT-fdeC* complex was the main virulence gene spectrum, accounting for 43.75% (7/16). It is widely believed that the increased serum survival (*iss*) gene was closely associated with the virulence of avian *E. coli* [32] and is a well-known virulence marker of extraintestinal pathogenic *E. coli* (ExPEC) in avian species, but this gene was not found in combination with other virulence markers of avian pathogenic *E. coli* (APEC) [33]. In this study, the *iss* gene was detected in three *E. coli* isolates (826red, 994, and 937), but no *eae* gene was detected. The three strains should be categorized among atypical ExPEC.

With the continuous development of sequencing technology, SGS technology has matured and is the best candidate technology in clinical settings, increasing the theoretical sensitivity of detection of various genes to provide higher coverage [34]. Single nucleotide polymorphism (SNP) analysis is a molecular marker technique developed for use with high-throughput sequencing [35] and as an indicator of the association between strains, can obtain more comprehensive and accurate results than MLST typing and is widely used in current research [36].

In the phylogenetic tree developed in this study, 16 endemic MDR isolates of *E. coli* from black-faced spoonbills were closely related to local strains in Liaoning Province, which indicated that the living or breeding habitat was the most important factor influencing the bacteria of migrating birds. Nine isolates were clustered in an independent branch and showed almost the same antimicrobial resistant phenotype, antimicrobial resistant genes, virulence genes, ST type, serotype and other characteristics, suggesting that they might have been obtained from one family of birds and that MDR *E. coli* was easily transferred among the members of a family. Strain 970 was closely related to strain112, isolated from Liaoning cows, and strain 937 was closely related to strain LNF11, isolated from Liaoning cattle feces, which suggests the possibility of cross infection between a small number of black-faced spoonbill populations and Liaoning cattle farms. Some researchers have proposed the necessity of strengthening the implementation of monitoring plans to detect the spread of zoonotic diseases between wild animals and humans and livestock [37]. The strains 958 and CX144T were both isolated from migratory birds and are in the same evolutionary branch, but the latter was reported in Zhejiang Province, which may indicate the risk of cross transmission of *E. coli* between the migration route of the black-faced spoonbill and the migration route of East Asian and Australian migratory birds.

## Conclusion

With the increasing maturity and decreasing cost of WGS, it can be more comprehensively and conveniently applied to molecular epidemiological research on MDR pathogens. The genetic diversity analysis showed that the *E. coli* isolates and their genetic elements carried by black-faced spoonbills on Shicheng Island might originate from their habitat and these strains could serve as a reservoir of resistance and virulence genes and spread among migratory birds. These multi-virulent and multi-resistant pathogens pose a potential threat to a rare species, the black-faced spoonbill, as well as to human public health. They should be monitored and evaluated regularly.

## Supporting information

**S1 File. Table S1**. Based on CLSI, the minimum inhibitory concentration (MIC) of 12 antibiotics for Enterobacteriaceae is shown with the concentration range of the E-test. **Table S2**. Primers for detecting virulence genes used in this study. **Table S3**. Accession numbers of the BioProject and BioSample for the genomes of 16 *E. coli* isolates. (DOCX)

## Acknowledgments

We are grateful to the members of the Wild Animal Sources and Diseases Inspection Station, National Forestry and Grassland Bureau of China, for help with the sampling.

## Author contributions

**Formal analysis:** Bing Liang, Bo-wen Jiang, Ling-wei Zhu, Jun Liu.

**Funding acquisition:** Yang Sun.

**Investigation:** Bing Liang, Bo-wen Jiang, Yue Yuan, Xue Ji.

**Project administration:** Yang Sun.

**Resources:** Tie-cheng Wang, Yuan-guo Li.

**Supervision:** Xue-jun Guo.

**Writing – original draft:** Bing Liang, Yue Yuan, Xue Ji.

**Writing – review & editing:** Yang Sun, Xue Ji.

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
