## [Decision Letter · Decision Letter 0]

22 Jan 2025

*Escherichia coli*

Dear Dr. Sun,

Thank you for submitting your manuscript to PLOS ONE. After careful consideration, we feel that it has merit but does not fully meet PLOS ONE’s publication criteria as it currently stands. Therefore, we invite you to submit a revised version of the manuscript that addresses the points raised during the review process.

We look forward to receiving your revised manuscript.

Kind regards,

Nabi Jomehzadeh, Ph.D (Assistant Professor)

Academic Editor

PLOS ONE

Journal Requirements:

5. Please include captions for your Supporting Information files at the end of your manuscript, and update any in-text citations to match accordingly. Please see our Supporting Information guidelines for more information: http://journals.plos.org/plosone/s/supporting-information .

Additional Editor Comments:

The manuscript offers an in-depth examination of the antimicrobial resistance and virulence factors associated with Escherichia coli strains found in black-faced spoonbills inhabiting Liaoning, China. The study clearly outlines its objectives, which include identifying specific resistance genes and assessing the potential health risks posed by these bacteria to both wildlife and humans.

Revise the abstract to emphasize the key findings and the broader implications derived from the study.

Enhance the discussion by offering a thorough overview of the study's limitations and any potential biases that may have influenced the results.

Focus on improving the clarity and visual presentation of the figures and tables to facilitate better understanding.

Expand the discussion section to incorporate a detailed analysis of the implications of the findings, as well as an exploration of the potential mechanisms that may explain the observed phenomena.

Finally, meticulously correct any minor errors present in the text, ensuring that all citations are accurate, properly formatted, and complete.

Reviewers' comments:

Reviewer's Responses to Questions

**Comments to the Author**

1. Is the manuscript technically sound, and do the data support the conclusions?

Reviewer #1: Partly

Reviewer #2: Yes

2. Has the statistical analysis been performed appropriately and rigorously?

Reviewer #1: Yes

Reviewer #2: Yes

3. Have the authors made all data underlying the findings in their manuscript fully available?

Reviewer #1: Yes

Reviewer #2: Yes

4. Is the manuscript presented in an intelligible fashion and written in standard English?

Reviewer #1: No

Reviewer #2: Yes

Reviewer #1: In general, this paper is interesting, and the authors have made an obvious effort. However, as a reviewer, I find this version of the paper requires more efforts to be valid to gain the attention of other researchers and readers. I have listed some points that could help in that manner (attached).

Reviewer #2: Dear researcher

Thanks for your nice work and interesting manuscript, all stages and processes were step-by-step and gentle, but some considerations exist in any study.

Please check these parts:

- Line 41, 43, 44, 47, 55, 74 each sentence in introduction need reference.

- Line 85 sample collection, clear the main method of fecal collection, the feces collected from the environment, bird, or etc.…?

- Line 97. Mention the incubator name and company and culture media reference number.

- Line 131. tete (M) change to tet.

- All references must be according to guidelines for authors and need for correction.

thank you, best regards.

**Do you want your identity to be public for this peer review?** For information about this choice, including consent withdrawal, please see our Privacy Policy

Reviewer #1: No

Reviewer #2: **Yes: ** Reza Nikzad

---

## [Author Response · Author response to Decision Letter 1]

7 Mar 2025

Dear Editors and Reviewers:

We would like to express our sincere gratitude to the editors and reviewers for their constructive comments on our paper (Submission ID:PONE-D-23-42911). These comments are the guidelines for us to improve our manuscript. After carefully reading the comments and suggestions, we have revised the manuscript that included the requested extra information. The revisions are marked in red in our revised manuscript and our point-by-point responses to the Journal and reviewers are as follows:

Responds to comments:

Journal Requirements:

Please ensure that your manuscript meets PLOS ONE's style requirements, including those for file naming. The PLOS ONE style templates can be found at https://journals.plos.org/plosone/s/file?id=wjVg/PLOSOne_formatting_sample_main_body.pdf

and

Response: Thank you for pointing this out, I modified the manuscript's style, according to the PLOS ONE's style requirements.

Response: Thank you for pointing this out, I ensure that the name of the authority that approved the field site access we were granted is the full name of the organization. As the sampling personnel are veterinary professionals, after professional training, they have the ability to collect fecal samples independently under the supervision of the Wild Animal Sources and Diseases Inspection Station, National Forestry and Grassland Bureau of China.

Response: Thank you for pointing this out, the Funding Information has been revised.

Response: Thank you for your constructive suggestion. When completing the data availability statement on the submission form, I did not take into account the publication timelines or the journal's open data policy. Here, I declare that all authors decide on a data sharing plan before acceptance.

Additional Editor Comments:

The manuscript offers an in-depth examination of the antimicrobial resistance and virulence factors associated with Escherichia coli strains found in black-faced spoonbills inhabiting Liaoning, China. The study clearly outlines its objectives, which include identifying specific resistance genes and assessing the potential health risks posed by these bacteria to both wildlife and humans.

1.Revise the abstract to emphasize the key findings and the broader implications derived from the study.

Response: Thank you for your constructive suggestion. Based on the suggestions of Reviewer 1, I have made revisions to the text. The final modification is as follows: In summary, black faced Spoonbill may become a reservoir of E. coli resistance and virulence genes, and interact with the ecological environment of its habitat during migration, seriously threatening the health of black faced Spoonbill and the public health and safety of its habitat.

2.Enhance the discussion by offering a thorough overview of the study's limitations and any potential biases that may have influenced the results.

Response: Thank you for your constructive suggestion. I will strengthen the impact of the limitations of this study on the results in the discussion section.

3.Focus on improving the clarity and visual presentation of the figures and tables to facilitate better understanding.

Response: Thank you for pointing this out, I will improve the clarity and visual presentation of the figures and tables.

4.Expand the discussion section to incorporate a detailed analysis of the implications of the findings, as well as an exploration of the potential mechanisms that may explain the observed phenomena.

Response: Thank you for your constructive suggestion. In this study, in the SNP evolutionary tree, 970 is closely related to 112 and 937 is closely related to LNF11, further verifying the convergence of gut microbiota and habitat microbiota in migratory birds, as well as the high plasticity of gut microbiota in ecological adaptation. I have supplemented this part in the discussion section.

5.Finally, meticulously correct any minor errors present in the text, ensuring that all citations are accurate, properly formatted, and complete.

Response: Thank you for pointing this out, I will seriously correct the small mistakes in the text to ensure that all quotations are accurate, correct and complete.

Reviewer #1

In general, this paper is interesting, and the authors have made an obvious effort. However, as a reviewer, I find this version of the paper requires more efforts to be valid to gain the attention of other researchers and readers. I have listed some points that could help in that manner.

1.Manuscript in general requires English writing enhancement

Response: Thank you for pointing this out, I will let a professional English native speaker with a doctor's degree polish the manuscript again.

Abstract

1.Line 20: This sentence is not clear:

“Seventy-four samples has E. coli isolated (29.7%), 43.2% of isolates carried an antimicrobial resistant phenotype and 21.6% isolates were considered to be MDR strains.”

Response: According to your valuable suggestion, I will revise this sentence to express its meaning clearly. Specifically revised as follows: “The results showed that the isolation rate of E. coli was 29.7% (74/249), with 43.2% (32/74) of isolates with drug-resistant phenotype and 21.6% (16/74) of isolates with multidrug resistance.”

2.Lines 30-35: This paragraph needs to be rewritten in a way that would highlight the importance and relevance of their findings to public health and to the reader's knowledge:

“In summary, E. coli may serve as a reservoir of resistance and virulence genes in migratory birds and may be transmitted to other species during migration, with virulent or multidrug resistant E. coli a potential threat to the rare black-faced spoonbill and to human public health”.

Response: According to your valuable suggestion, I will further modify the conclusion as follows:

In summary, black faced Spoonbill may become a reservoir of Escherichia coli resistance and virulence genes, and interact with the ecological environment of its habitat during migration, seriously threatening the health of black faced Spoonbill and the public health and safety of its habitat.

Introduction

1.The introduction should describe the main pathogen first (E. coli) and its importance as a main reservoir for transmission of ARGs to other pathogens. Then, a short but informative few paragraphs about migratory birds and black-faced spoonbills would provide a strong introduction to the proposed role of black-faced spoonbills in spreading pathogens and antibiotic-associated genes.

Response: Thank you for your constructive suggestion. I will readjust the description of the introduction.

Materials and methods

Over all, the authors clearly put in a lot of efforts to finalise this paper. However, crucial questions have been raised about their methodologies.

1: Using boiling method for DNA extraction:

For some researchers, it is accepted once the boiling method is used for bacterial DNA extraction. It would provide a low-cost method. However, it may provide low-quality DNA that may affect the results of the detection of virulence and drug-resistant associated genes.

Response: Thank you for your constructive suggestion. Although this method for extracting DNA has certain drawbacks, it is feasible for epidemiological studies involving a large number of isolates, as listed in the references. Although the quality of DNA extracted by this method is not as good as that extracted by the reagent kit, it is suitable for PCR identification of pure isolated cultures.

1.Elbediwi M, Shi D, Biswas S, Xu X, Yue M. Changing Patterns of Salmonella enterica Serovar Rissen From Humans, Food Animals, and Animal-Derived Foods in China, 1995-2019. Front Microbiol. 2021 Jul 29;12:702909. doi: 10.3389/fmicb.2021.702909[frontiers in microbiology]

2.Carrasco G, Valdezate S, Garrido N, Villalón P, Medina-Pascual MJ, Sáez-Nieto JA. Identification, typing, and phylogenetic relationships of the main clinical Nocardia species in spain according to their gyrB and rpoB genes. J Clin Microbiol. 2013 Nov;51(11):3602-8. doi: 10.1128/JCM.00515-13[journal of clinical microbiology]

3.Salah FD, Soubeiga ST, Ouattara AK, Sadji AY, Metuor-Dabire A, Obiri-Yeboah D, Banla-Kere A, Karou S, Simpore J. Distribution of quinolone resistance gene (qnr) in ESBL-producing Escherichia coli and Klebsiella spp. in Lomé, Togo. Antimicrob Resist Infect Control. 2019 Jun 18;8:104. doi: 10.1186/s13756-019-0552-0[antimicrobial resistance and infection control]

4.Ibrahim DR, Dodd CER, Stekel DJ, Meshioye RT, Diggle M, Lister M, Hobman JL. Multidrug-Resistant ESBL-Producing E. coli in Clinical Samples from the UK. Antibiotics (Basel). 2023 Jan 13;12(1):169. doi: 10.3390/antibiotics12010169[ANTIBIOTICS BASEL ]

5.Liu Q, Ma A, Wei L, Pang Y, Wu B, Luo T, Zhou Y, Zheng HX, Jiang Q, Gan M, Zuo T, Liu M, Yang C, Jin L, Comas I, Gagneux S, Zhao Y, Pepperell CS, Gao Q. China's tuberculosis epidemic stems from historical expansion of four strains of Mycobacterium tuberculosis. Nat Ecol Evol. 2018 Dec;2(12):1982-1992. doi: 10.1038/s41559-018-0680-6[ Nature ecology & evolution ]

6.Shen Y, Zhou H, Xu J, Wang Y, Zhang Q, Walsh TR, Shao B, Wu C, Hu Y, Yang L, Shen Z, Wu Z, Sun Q, Ou Y, Wang Y, Wang S, Wu Y, Cai C, Li J, Shen J, Zhang R, Wang Y. Anthropogenic and environmental factors associated with high incidence of mcr-1 carriage in humans across China. Nat Microbiol. 2018 Sep;3(9):1054-1062. doi: 10.1038/s41564-018-0205-8[Nature microbiology]

7.Yuan Y, Liang B, Jiang BW, Zhu LW, Wang TC, Li YG, Liu J, Guo XJ, Ji X, Sun Y. Migratory wild birds carrying multidrug-resistant Escherichia coli as potential transmitters of antimicrobial resistance in China. PLoS One. 2021 Dec 15;16(12):e0261444. doi: 10.1371/journal.pone.0261444[PLOS ONE]

2: Using an automated microbial and antimicrobial identification system

It is well known (the authors have also stated that clearly in their discussion) that the commercially available automated microbial and antimicrobial identification systems are providing results with variable and inconsistent accuracies. Therefore, authors should describe the obtained accuracy rate following using that system for identification of E. coli.

Response: Thank you for your constructive suggestion. I have supplemented the accuracy in the text. The supplementary content is as follows: According to the BD Phoenix TM-100 automated microbial identification system, the bacteria has been confirmed as E. coli with over 99% similarity.

3: The authors decided to use conventional PCR assay to detect virulence and antibiotic resistant associated genes in the first place, then they moved to use SGS assay to screen only 16 isolates under just one term which is the MDR phenotype.

This provides an impression about that the authors might not be quite sure about the primary results of E. coli identification and MDR phenotypes. Therefore, it would be a good suggestion if the authors would change the story of this paper to describe the conventional methods at first and then describe the addition of the SGS assay to unravelling the mysteries behind some inconsistent results.

Response: Thank you for pointing out this point. In the article, we first performed PCR detection on the major virulence and resistance genes, with the aim of using them as screening markers to screen all isolates and ultimately selecting 16 strains of Escherichia coli for SGS.

4: Why have the authors were decided to use SGS over NGS or WGS? The SGS system has some issues with sequence depths that may affect the perfect annotations and sequence closing.

Response: Thank you for raising this question. After careful consideration, we believe that although SGS does not have the depth of NGS or WGS in reading sequences, it can accurately detect most of the resistance genes and virulence genes. Secondly, considering the overall funding, we have chosen SGS.

Results

Visualise the phylogenetic tree of the 16 isolates with adding the host and sample location to each isolate was very nice and informative. It is a good choice from the authors.

It seems that isolates number 937 and 970 were the only two significant isolates that have been chosen to be sequenced so far. They share significant similarity and close relationship with two other isolates from cattle and Mink, respectively. The authors are encouraged to pay more effort to generate genome comparison between isolate 937 and LNF11 and between isolate 970 and 112 as well. This would make the story more interesting.

Response: Thank you for your valuable suggestion. In this study, the SNP evolutionary tree revealed a significant correlation between strain 937 and strains LNF11, strain 970 and strain 112. However, through cgMLST comparison, significant allelic differences were found between strain 937 and the strains LNF11, strain 970 and strain 112. We plan to conduct in-depth research on the genomic relevance between strain 937 and the strains LNF11, strain 970 and strain 112 in future studies.

Discussion

1: The discussion clearly showed that the authors were not biased about their methodology and findings.As mentioned earlier (section 3 – Material and Methods), the discussion section needs to be rewritten in a more confident way about the results. The current version of discussion provides an impression of uncertainty about the results.

Response: Thank you for your valuable feedback. I will revise the discussion section to ensure it is presented with greater confidence and to provide a more definitive analysis of the discussion's outcomes where possible.

2: In line 332, the authors have mentioned that strain 112 was isolated from Liaoning cows but in the figure of phylogenetic tree it was noted as isolated from Mink. They need to correct that using the proper source please.

Response: I have confirmed that the isolate 112 was obtained from mink in Liaoning Province, and I have made the necessary modifications in this paper.

3: In line 336: The authors mentioned some relationship between strains 958 and CX144T due to their localisation in a same branch. The phylogenetic tree does not imply this statement.

Response: Thank you for your constructive suggestion. Due to my oversight, 958 and CX144T were mistakenly classified as being on the same evolutionary branch. I have removed the discussion section pertaining to 958 and CX144T. I am deeply sorry for any inconvenience this may have caused you.

Reviewer #2:

Please check these parts:

1.- Line 41, 43, 44, 47, 55, 74 each sentence in introduction need reference.

Response: Thank you for your valuable suggestion. Although the "Introduction" paragraph has been revised, I have added references where requested.

2.- Line 85 sample collection, clear the main method of fecal collect

---

## [Decision Letter · Decision Letter 1]

22 Jun 2025

Dear Dr. Sun,

Thank you for submitting your manuscript to PLOS ONE. After careful consideration, we feel that it has merit but does not fully meet PLOS ONE’s publication criteria as it currently stands. Therefore, we invite you to submit a revised version of the manuscript that addresses the points raised during the review process.

We look forward to receiving your revised manuscript.

Kind regards,

Nabi Jomehzadeh, Ph.D (Assistant Professor)

Academic Editor

PLOS ONE

Journal Requirements:

Reviewers' comments:

Reviewer's Responses to Questions

**Comments to the Author**

Reviewer #2: All comments have been addressed

Reviewer #3: (No Response)

2. Is the manuscript technically sound, and do the data support the conclusions?

Reviewer #2: Yes

Reviewer #3: Yes

3. Has the statistical analysis been performed appropriately and rigorously?

Reviewer #2: Yes

Reviewer #3: Yes

4. Have the authors made all data underlying the findings in their manuscript fully available?

Reviewer #2: Yes

Reviewer #3: Yes

5. Is the manuscript presented in an intelligible fashion and written in standard English?

Reviewer #2: Yes

Reviewer #3: Yes

Reviewer #2: (No Response)

Reviewer #3: Introduction

Lines 48 and 50: Are these dates correct? (1830, 1880 and 1989?)

Lines 54 and 55: This sentence needs a reference

Lines 73 and 74: This sentence also needs a reference.

Lines 78 to 83: This paragraph should be rewritten with a focus on the objective of the work. The way it is written, it is quite confusing and does not make the objective of the work clear. I suggest starting with the research of the microorganism in the feces and then the tests that will be performed.

General comment on the introduction:

The introduction of the work is extensive, with a broad revision. In principle, I do not see a problem with this. The final paragraph should be rewritten with a focus on the objectives, and should be written in the sequential order of the research.

Material and methods

Sample collection

The collection of the sample should be mentioned, whether it was collected in the environment or if it was handled by the animal, and in this case, there should be some document from the animal ethics committee. It is assumed that it was collected by handling the animal, since it is mentioned that it did not “cause harm to the animal”. This must be documented through official authorization, either by the institution carrying out the research or by environmental agencies.

Bacterial isolation

The reference to the microbial isolation protocol and the procedures (step by step) for microbiological isolation were not mentioned.

Antimicrobial susceptibility test

The experimental protocol was missing. Was the one recommended by the equipment used? It must be mentioned.

DNA extraction

It must be written that the supernatant was used as a template

Results

No comments.

Discussion

Line 281: include the observation by the authors Yuan Yue´s (25) to corroborate the results of the study.

Lines 295 to 297: Could the lack of detection of genes related to chloramphenicol be related to some loss in the trimming?

Lines 301 to 306: This discussion between the comparison between the two techniques of PCR and SGS does not seem appropriate to me, given the finding of the presence of the chloramphenicol resistance gene, as well as the high MIC value and non-detection in SGS. Taking into account the costs between one and the other, perhaps a discussion addressing the uses of the tools in a collaborative and not antagonistic way would be more appropriate, taking into account the flaws that SGS can present.

General comment

The work is relevant because it verifies the presence of Escherichia coli with virulence factors and antimicrobial resistance in the black-faced spoonbill (Platalea minor), which in turn is in danger of extinction. The results contribute to the monitoring of the microorganism in relation to pathogenicity and antimicrobial resistance in the world.

**Do you want your identity to be public for this peer review?** For information about this choice, including consent withdrawal, please see our Privacy Policy

Reviewer #2: No

Reviewer #3: **Yes: ** Angela Patricia Santana

---

## [Author Response · Author response to Decision Letter 2]

11 Nov 2025

Dear Editors and Reviewers:

We would like to express our sincere gratitude to the editors and reviewers for their constructive comments on our paper (Submission ID:PONE-D-23-42911). These comments are the guidelines for us to improve our manuscript. After carefully reading the comments and suggestions, we have revised the manuscript that included the requested extra information. The revisions are marked in red in our revised manuscript and our point-by-point responses to the Journal and reviewers are as follows:

Responds to comments:

Journal Requirements:

Response: Thank you for pointing this out, I have checked the list of references and confirmed their completeness and correctness. The revised references have been marked in the revised manuscript with track changes.

2. Lines 48 and 50: Are these dates correct? (1830, 1880 and 1989?)

Response: Thank you for pointing this out, but in this article, the 1830s refers to the 30s of the 19th century; the 1880s refers to the 80s of the 19th century; and 1989 is the numerical representation of the year nineteen hundred and eighty-nine.

3. Lines 54 and 55: This sentence needs a reference

Response: Thank you for your valuable suggestion. I have added references where requested.

4. Lines 73 and 74: This sentence also needs a reference.

Response: Thank you for your valuable suggestion. I have added references where requested.

5. Lines 78 to 83: This paragraph should be rewritten with a focus on the objective of the work. The way it is written, it is quite confusing and does not make the objective of the work clear. I suggest starting with the research of the microorganism in the feces and then the tests that will be performed.

Response: Thank you for your constructive suggestion. I will readjust the expression of the last paragraph of the introduction. The readjust content is as follows:

This study systematically analyzes E. coli strains isolated from black-faced spoonbills (Platalea minor) in Liaoning Province, China. We aim to (1) comprehensively characterize the virulence factor profiles and antimicrobial resistance spectra of these isolates, (2) investigate the distribution patterns of integrase genes and their potential associations with resistance genes, and (3) employ whole-genome single-nucleotide polymorphism (SNP) analysis to assess genetic relatedness among strains. These analyses will elucidate the transmission dynamics and ecological implications of E. coli carried by this avian host, providing insights into its public health and environmental significance.

6. The introduction of the work is extensive, with a broad revision. In principle, I do not see a problem with this. The final paragraph should be rewritten with a focus on the objectives, and should be written in the sequential order of the research.

Response: Thank you for your constructive suggestion. We have re-described the last paragraph of the introduction, with the focus of the description being primarily on the research objectives.

7. The collection of the sample should be mentioned, whether it was collected in the environment or if it was handled by the animal, and in this case, there should be some document from the animal ethics committee. It is assumed that it was collected by handling the animal, since it is mentioned that it did not “cause harm to the animal”. This must be documented through official authorization, either by the institution carrying out the research or by environmental agencies.

Response: Thank you for your constructive suggestion. I have included in the article the approval from the Experimental Animal Welfare and Ethics Committee of Changchun Veterinary Research Institute, Chinese Academy of Agricultural Sciences, with the approval number: AMMS-11-2020-11.

8.The reference to the microbial isolation protocol and the procedures (step by step) for microbiological isolation were not mentioned.

Response: Thank you for your constructive suggestion. This study does not mention the microbial isolation protocol nor describe the procedures of microbial isolation step by step. However, the isolation and identification of Escherichia coli in this study adopted conventional isolation methods, and several references are provided here for corroboration.

[1] Wibisono FJ, Effendi MH, Tyasningsih W, Rahmaniar RP, Khairullah AR, Kendek IA, Budiastuti B, Rianto V, Nico DC, Kurniasih DAA, Salwa S, Diningrum DP, Moses IB, Ahmad RZ. Antibiotic resistance profiles of Escherichia coli and Salmonella spp. isolated from chicken meat sold in traditional markets in Gresik District, East Java, Indonesia. Open Vet J. 2025 May;15(5):2160-2170. doi: 10.5455/OVJ.2025.v15.i5.34.

[2] Sharew SG, Weldehanna DG, Gebreyes DS, Sahile Z, Abebe TA, Shibabaw A. Extended spectrum betalactamase and carbapenemase producing gram negative bacteria from mobile phones of healthcare workers at Debre Berhan Hospital, Ethiopia. Sci Rep. 2025 May 26;15(1):18427. doi: 10.1038/s41598-025-03191-5.

[3] Byarugaba I, Nabatanzi A, Muhumuza E, Kyambadde J. Impact of heavy metals on antibiotic resistance of Escherichia coli from slum wastewater in Kawempe division, Kampala district, Uganda: a case study. BMC Microbiol. 2025 May 21;25(1):310. doi: 10.1186/s12866-025-04024-1.

[4] Al-Groom R. Incidence of extended spectrum beta-lactamase (ESBL) producing Escherichia coli isolated from women with urinary tract infections in Jordan. Iran J Microbiol. 2025 Feb;17(1):41-50. doi: 10.18502/ijm.v17i1.17800.

9. The experimental protocol was missing. Was the one recommended by the equipment used? It must be mentioned.

Response: It was recommended by the equipment used. The BD Phoenix™-100 automated microbial identification system mentioned in this study complies with a set of antimicrobial susceptibility testing established by CLSI, and several references are provided here for corroboration.

[1] Wang G, Zhu Y, Feng S, Wei B, Zhang Y, Wang J, Huang S, Qin S, Liu X, Chen B, Cui W. Extended-spectrum beta-lactamase-producing Enterobacteriaceae related urinary tract infection in adult cancer patients: a multicenter retrospective study, 2015-2019. BMC Infect Dis. 2023 Mar 6;23(1):129. doi: 10.1186/s12879-023-08023-3.

[2] Jian X, Du S, Zhou X, Xu Z, Wang K, Dong X, Hu J, Wang H. Development and validation of nomograms for predicting the risk probability of carbapenem resistance and 28-day all-cause mortality in gram-negative bacteremia among patients with hematological diseases. Front Cell Infect Microbiol. 2023 Jan 5;12:969117. doi: 10.3389/fcimb.2022.969117.

[3] Stanley IJ, Kajumbula H, Bazira J, Kansiime C, Rwego IB, Asiimwe BB. Multidrug resistance among Escherichia coli and Klebsiella pneumoniae carried in the gut of out-patients from pastoralist communities of Kasese district, Uganda. PLoS One. 2018 Jul 17;13(7):e0200093. doi: 10.1371/journal.pone.0200093.

[4] Liu K, Xu H, Sun J, Liu Y, Li W. Investigation and analysis of carbapenem-resistant gram-negative bacterial infection rates across hospitals in Shandong Province in China. Front Public Health. 2022 Nov 7;10:1014995. doi: 10.3389/fpubh.2022.1014995.

10. It must be written that the supernatant was used as a template and several references are provided here for corroboration.

Response: According to your valuable suggestion, I have specifically stated in the article that the supernatant was used as a template.

11. Line 281: include the observation by the authors Yuan Yue´s (25) to corroborate the results of the study.

Response: Thank you for your constructive suggestion. I've already refined the sentence based on your reminders.

12. Lines 295 to 297: Could the lack of detection of genes related to chloramphenicol be related to some loss in the trimming?

Response: Thank you for your constructive suggestion. Thank you for your valuable suggestions. In addition to the reasons analyzed in the text, the read length of next-generation sequencing (NGS) is typically around several hundred base pairs, which may lead to sequence bias. Moreover, since next-generation sequencing primarily relies on PCR amplification, it is characterized by long sequencing time, short read length, and poor end-sequence quality. All these factors may affect the detection results.

13. Lines 301 to 306: This discussion between the comparison between the two techniques of PCR and SGS does not seem appropriate to me, given the finding of the presence of the chloramphenicol resistance gene, as well as the high MIC value and non-detection in SGS. Taking into account the costs between one and the other, perhaps a discussion addressing the uses of the tools in a collaborative and not antagonistic way would be more appropriate, taking into account the flaws that SGS can present.

Response: Thank you for your constructive suggestion. The necessary modifications have been made in the text.

14. The work is relevant because it verifies the presence of Escherichia coli with virulence factors and antimicrobial resistance in the black-faced spoonbill (Platalea minor), which in turn is in danger of extinction. The results contribute to the monitoring of the microorganism in relation to pathogenicity and antimicrobial resistance in the world.

Response: Thank you for your recognition of this research.

Response: Thank you for pointing out this point. I have uploaded the image to the Preflight Analysis and Conversion Engine (PACE) and downloaded the graphics that meet PLOS's requirements.

---

## [Decision Letter · Decision Letter 2]

7 Dec 2025

Antimicrobial resistance and virulence analysis of Escherichia coli carried by black-faced spoonbill (Platalea minor) in Liaoning, China

PONE-D-23-42911R2

Dear Dr. Sun,

We’re pleased to inform you that your manuscript has been judged scientifically suitable for publication and will be formally accepted for publication once it meets all outstanding technical requirements.

Kind regards,

Nabi Jomehzadeh, Ph.D (Associate Professor)

Academic Editor

PLOS One

Additional Editor Comments (optional):

Reviewers' comments:

Reviewer's Responses to Questions

**Comments to the Author**

Reviewer #3: All comments have been addressed

2. Is the manuscript technically sound, and do the data support the conclusions?

Reviewer #3: Yes

3. Has the statistical analysis been performed appropriately and rigorously?

Reviewer #3: Yes

4. Have the authors made all data underlying the findings in their manuscript fully available?

Reviewer #3: Yes

5. Is the manuscript presented in an intelligible fashion and written in standard English?

Reviewer #3: Yes

Reviewer #3: The suggestions for corrections, as well as the inclusion of bibliographic references, were accepted; therefore, I consider the article to be suitable for publication.

**Do you want your identity to be public for this peer review?** For information about this choice, including consent withdrawal, please see our Privacy Policy

Reviewer #3: **Yes: ** Angela Patricia Santana

---

## [Editor Report · Acceptance letter]

PONE-D-23-42911R2

PLOS One

Dear Dr. Sun,

I'm pleased to inform you that your manuscript has been deemed suitable for publication in PLOS One. Congratulations! Your manuscript is now being handed over to our production team.

Kind regards,

on behalf of

Dr. Nabi Jomehzadeh

Academic Editor

PLOS One